# Facile Fabrication of α-Bisabolol Nanoparticles with Improved Antioxidant and Antibacterial Effects

**DOI:** 10.3390/antiox12010207

**Published:** 2023-01-16

**Authors:** Sangwoo Kim, Sohyeon Yu, Jisu Kim, Nisar Ul Khaliq, Won Il Choi, Hyungjun Kim, Daekyung Sung

**Affiliations:** 1Center for Bio-Healthcare Materials, Bio-Convergence Materials R&D Division, Korea Institute of Ceramic Engineering and Technology, 202 Osongsaengmyeong 1-ro, Osong-eup, Heungdeok-gu, Cheongju 28160, Republic of Korea; 2Department of Chemical and Biomolecular Engineering, Yonsei University, 50 Yonsei-ro, Seodaemun-gu, Seoul 03722, Republic of Korea; 3Department of Chemical Engineering, Hanyang University, 222 Wangsimni-ro, Seongdong-gu, Seoul 04763, Republic of Korea; 4Department of Chemistry and Bioscience, Kumoh National Institute of Technology, 61 Daehak-ro, Gumi 39177, Republic of Korea

**Keywords:** bioactive compound, α-bisabolol, polyglyceryl-4 caprate, nanoparticle, solubilization, encapsulation, antioxidant, antibacterial

## Abstract

Bioactive compounds are widely used in the bio-industry because of their antioxidant and antibacterial activities. Because of excessive oxidative stress, which causes various diseases in humans, and because preservatives used in bioproducts cause allergies and contact dermatitis, it is important to use natural bioactive compounds in bioproducts to minimize oxidative stress. α-bisabolol (ABS) is a natural compound with both antioxidant and antibacterial properties. However, its water-insolubility makes its utilization in bioproducts difficult. In this study, ABS-loaded polyglyceryl-4 caprate nanoparticles (ABS@NPs) with improved aqueous stability and ABS loading were fabricated using an encapsulation method. The long-term stability of the ABS@NPs was analyzed with dynamic light scattering and methylene blue-staining to determine the optimized ABS concentration in ABS@NPs (10 wt%). The ABS@NPs exhibited excellent antioxidant activity, according to the 2,2-diphenyl-1-picrylhydrazyl assay and in vitro reactive oxygen species generation in NIH-3T3 fibroblast cells, and an outstanding antibacterial effect, as determined using the *Staphylococcus aureus* colony-counting method. Furthermore, we evaluated the biocompatibility of the ABS@NPs in vitro. This study suggests that ABS@NPs with improved antioxidant and antibacterial properties can be used to treat diseases related to various oxidative stresses and can be applied in many fields, such as pharmaceuticals, cosmetics, and foods.

## 1. Introduction

Bioactive compounds have various properties, such as antioxidant and antibacterial activities, and they are widely used in the pharmaceutical, cosmetic, and food industries [1,2,3]. Terpenoid, a representative bioactive compound, is a natural organic compound composed of isoprene units that is produced by isopentenyl diphosphate, which is a precursor to isoprene, and its isomer, dimethylallyl diphosphate (DMAPP). It is known for its excellent ability to scavenge reactive oxygen species (ROS), that is, antioxidant ability, owing to which it has been a subject of many studies [4,5,6]. ROS, which are highly reactive forms of intermediates, include hydrogen peroxide (H_2_O_2_), superoxide radical (O_2_^−^), and hydroxyl radical (OH^−^). They cause extreme damage to biomolecules in cells and mitochondria, thereby inducing inflammation and excessive oxidative stress in the body resulting in various diseases, such as cancer and Alzheimer’s disease (AD), in humans [7,8,9,10,11,12]. Therefore, compounds with antioxidant properties that have the ability to remove these free radicals have garnered considerable interest in relation to the development of various bioproducts in the modern urbanized society, where many factors, such as ultraviolet rays, fine dust, and blue light, induce the formation of active oxygen species [13,14,15].

Antibacterial activity is an important property for bioproduct development [16]. Compared to chemical and plastic products, bioproducts are more susceptible to contamination and denaturation, which makes it difficult to maintain the product quality [16,17,18,19]. As bioproducts are in an aqueous environment, where microorganisms, such as bacteria, can easily grow, it is essential to add preservatives to prevent their degradation [19,20,21]. In the bio-industry, preservatives are added to decrease the bacteria purity during manufacturing, packaging, and storage [21]. However, preservatives such as formaldehyde, iodopropynyl butylcarbamate, and triclosan cause allergies and contact dermatitis in users, even when used in small amounts; therefore, their use is being reduced gradually [22,23,24]. One way to minimize this issue is to use natural substances with antibacterial effects in bioproducts [25,26,27].

In this study, we focused on α-bisabolol (ABS), a natural bioactive compound with antioxidant and antibacterial properties. ABS is a naturally occurring sesquiterpene alcohol that is widely present in plant sources, such as *Matricaria chamomilla*, and several studies have indicated that it has useful pharmacological and cosmetic activities, including anti-inflammatory, anti-irritant, anti-tumor, antioxidant, antibacterial, and anti-allergic properties [28,29,30,31,32,33]. However, owing to the insolubility of ABS in water, its bioavailability is low, and it is difficult to develop a bioproduct based on ABS due to its lipophilicity [28,34,35,36,37]. To overcome these disadvantages, nanocapsule techniques using amphiphilic polymers or solubilizers have been developed to enhance the bioavailability and solubility of hydrophobic substances, such as ABS [7,28,29,30,36,37]. However, the ABS loading and physicochemical stability of the previously developed systems are low, the manufacturing process is complicated, and the fabrication requires temperature-controlled environmental conditions [29,30,31,37]. Therefore, the development of an ideal nanocarrier for ABS requires an optimization technology that can enhance both the loading and the physicochemical stability of ABS in a simple and convenient manner.

In this study, we developed a nanoprecipitation technique using amphiphilic polyglyceryl-4 caprate, which contains a hydrophilic carboxyl group segment and a hydrophobic hydrocarbon chain segment, to fabricate ABS-loaded nanoparticles (ABS@NPs) with higher loading and stability of ABS than previously achieved. The objective of this study was to improve the long-term stability and effectiveness of ABS as an antioxidant and antibacterial material in an aqueous environment (Figure 1). To evaluate the stability of ABS@NPs, many physicochemical properties, such as the particle diameter, surface charge, and polydispersity index (PDI), were analyzed via dynamic light scattering (DLS). The stability of the nanoparticle system was also confirmed visually using a methylene blue staining test. The ROS-scavenging activity of ABS@NPs was analyzed in situ using the 2,2-diphenyl-1-picrylhydrazyl (DPPH) assay and through in vitro studies using mouse embryonic fibroblast cells (NIH 3T3). Furthermore, the antibacterial activity of ABS@NPs was confirmed using the colony-counting method. Finally, we confirmed the biocompatibility of ABS@NPs with an in vitro experiment using NIH 3T3 cells. The improved antioxidant and antibacterial properties of ABS@NPs make them suitable candidates for treating diseases related to oxidative stress, and they are expected to have applications in various fields, such as pharmaceuticals, cosmetics, and foods.

## 2. Materials and Methods

### 2.1. Materials

ABS and tetrahydrofuran (THF; anhydrous, 99.9%) were purchased from Sigma-Aldrich (St. Louis, MO, USA), polyglyceryl-4 caprate was purchased from Special Chemical Co. (Paris, France), and deionized water (DIW) were obtained from HyClone (Logan, UT, USA) for fabricating ABS nanoparticles. For high-performance liquid chromatography (HPLC), deionized water (DIW) was purchased from Milli-Q^®^ (Molsheim, France), and acetonitrile (HPLC grade) and phosphoric acid were obtained from Merck^®^ (Darmstadt, Germany). For the dye solubility method, methylene blue was obtained from Sigma-Aldrich (St. Louis, MO, USA). For the in situ antioxidants experiment, the reagents 2,2-diphenyl-1-picrylhydrazyl (DPPH) and ascorbic acid (AA) were obtained from Sigma-Aldrich (St. Louis, MO, USA). For in vitro cell incubation, penicillin–streptomycin, trypsin (0.25%), fetal bovine serum (FBS), and dulbecco’s modified Eagle’s medium (DMEM) were purchased from Gibco (Grand Island, NY, USA). For in vitro experiments, reagents such as 2,7-dichlorodihydrofluorescein diacetate (H2DCFDA) and 3-(4,5-dimethylthiazol-2-yl)-2,5-diphenyltetrazolium bromide (MTT) were obtained from Invitrogen (Carlsbad, CA, USA), hydrogen peroxide (H_2_O_2_, 30%) was purchased from Junsei Chemical Co. (Tokyo, Japan), dimethyl sulfoxide-d6 (DMSO-d6, 99.8%) was obtained from Sigma-Aldrich (St. Louis, MO, USA), and phosphate-buffered saline (PBS, pH 7.4) was purchased from HyClone (Logan, UT, USA). For the Gram-positive bacterium, *Staphylococcus aureus* ATCC 6538, culture, Luria–Bertani (LB) agar and Luria–Bertani (LB) broth were obtained from BD Difco (Sparks, MD, USA). For the antibacterial assay, the Gram-positive bacterium, *Staphylococcus aureus* ATCC 6538 was obtained from the American Type Culture Collection (ATCC; Manassas, VA, USA). All solvents were used as received without additional purification.

### 2.2. Preparation of ABS@NPs

ABS@NPs were fabricated through self-assembly in the aqueous phase after mixing hydrophobic ABS with amphiphilic polyglyceryl-4 caprate, which contains a hydrophilic carboxyl group segment and a hydrophobic hydrocarbon chain segment. First, ABS (1, 2, 4, or 8 mg) and polyglyceryl-4 caprate (20 mg) were dissolved in THF (1 mL) in 4 mL vial under magnetic stirring at room temperature for 2 h. Then, utilizing a syringe pump (LEGATO100, KD Scientific, Korea), the THF solution with ABS and polyglyceryl-4 caprate was added dropwise to 5 mL of DIW with stirring at 530 rpm. The ABS@NPs (ABS loading of 5, 10, 20, and 40 wt% with respect to polyglyceryl-4 caprate) were stabilized by mild stirring (530 rpm) for 1 h. Finally, the solvent was completely evaporated via vacuum evaporation for 2 h and the remaining was diluted with DIW to 5 mL. The resulting products are denoted as ABS@NP × wt%, where × wt% represents the ABS loading respect to polyglyceryl-4 caprate. The same method was used to prepare empty nanoparticles (NPs) of polyglyceryl-4 caprate without the addition of ABS (denoted as ABS@NP 0 wt%) [7]. The zeta potential, PDI, and particle size of ABS@NPs were estimated through DLS using an electrophoretic light-scattering spectrophotometer (ELS-Z2, Otsuka Electronics Co., Tokyo, Japan). The loading content and loading efficiency of ABS in nanoparticles were analyzed via HPLC of unloaded ABS. Utilizing Amicon Ultra-15 centrifugal filters (molecular weight cutoff: 100 kDa), unloaded ABS was purified through ultrafiltration at 2,000 rpm for 15 min [36] The chromatographic conditions used for the HPLC experiment are as follows: the injection volume was 20 μL, the detection wavelength was 200 nm, and the mobile phases were a mixture of (A) acetonitrile and (B) water. In addition, acetonitrile and phosphoric acid were used at 80:19:1 by volume in a linear gradient elution program from A:B (50:50) to 100% A in 25 min, returning to A:B (50:50) in 5 min. The eluent flow rate was 0.8 mL/min. All HPLC experiments were performed at room temperature. After this, the loading content and loading efficiency were calculated as follows [34,35,36]:(1)$$Loading\;content\;\left(\%\right)=\left[\frac{\left(weight\;of\;fed\;ABS-weight\;of\;unloaded\;ABS\right)}{weight\;of\;NPs}\right]\times 100$$
(2)$$Loading\;efficiency\;\left(\%\right)=\left[\frac{\left(weight\;of\;fed\;ABS-weight\;of\;unloaded\;ABS\right)}{weight\;of\;fed\;ABS}\right]\times 100$$

### 2.3. Methylene Blue Staining of ABS@NPs

To visually confirm the encapsulation, stability, and transparency of the dispersed phase of the fabricated ABS@NPs with various ABS concentrations (0 wt%, 5 wt%, 10 wt%, 20 wt%, and 40 wt%), a dye solubility method was carried out utilizing methylene blue, a hydrophilic dye. First, methylene blue was dissolved in DIW at 120 nM concentration. Next, 2 mL of the dye solution was added to 2 mL of the ABS@NPs suspension and mixed in a 4 mL vial. Finally, the solution was visually examined to confirm the precipitation or dispersion of ABS@NPs [7,38].

### 2.4. Assessment of the Stability of ABS@NPs

The stability of ABS@NPs was analyzed over 112 days via DLS. The PDI and particle size of ABS@NPs were estimated at ABS concentrations of 0, 5, and 10 wt% in DIW on days 0, 1, 3, 7, 14, 28, 56, and 112.

### 2.5. In Situ Antioxidant Activity of ABS@NPs by the DPPH Assay

The antioxidant activity of ABS@NPs was evaluated utilizing a DPPH assay [39]. As control groups, solutions of AA and ABS in DIW were prepared at two concentrations each (5 and 10 wt%). The ABS@NPs were prepared at three ABS concentrations (0, 5, and 10 wt%). Then, a 0.1 mM DPPH solution was diluted with ethanol and stored in dark conditions at 4 °C. Subsequently, the prepared DPPH solution 50 µL was added with each ABS@NP suspension of 150 µL. The negative control group was 50 µL of DPPH solution added to 150 µL of DIW, which hardly showed any antioxidant activity. As a positive control, AA was used and ABS in DIW and ABS@NP 0 wt% were used as negative control groups for comparison with ABS@NPs. All mixtures were stored in dark condition at 25 °C for 24 h. Using a microplate reader (VICTOR X5, PerkinElmer, Singapore, Republic of Singapore), the absorbance of each mixture was analyzed at a wavelength of 515 nm. The antioxidant activity was calculated utilizing the following equation: [40,41]
(3)$$Antioxidant\;activity\;\left(\%\right)=\left[\frac{\Delta A_{515}\;of\;control-\Delta A_{515}\;of\;sample}{\Delta A_{515}\;of\;control}\right]\times 100$$

### 2.6. In Vitro Antioxidant Activity of ABS@NPs

To estimate the intracellular antioxidant effect of the ABS@NPs with ABS concentrations of 5 and 10 wt%, mouse embryonic fibroblast cells (NIH 3T3) were cultured in 96-well plates at a density of 10,000 cells/well and incubated for 24 h. ROS were induced in NIH 3T3 fibroblast cells via stimulation with H_2_O_2_, as an oxidative stress agent. After sample treatment, changes in the ROS levels were estimated using the following steps: first, suspensions of ABS@NP 10 wt% with various ABS concentrations (1–1000 nM) and 5 μM hydrogen peroxide (H_2_O_2_) were added to fibroblast cells and cultured for 8 h. For comparison, a negative control group without H_2_O_2_ and a positive control group with H_2_O_2_ were used. Then, after washing with PBS (200 µL to each 96-well plate) to eliminate the remaining sample solution, the NIH 3T3 cells were treated with a 10 μM H2DCFDA solution, which is an ROS fluorescence indicator, and seeded for 90 min in the dark. Finally, the in vitro antioxidant activity of the samples was estimated by measuring the fluorescence intensity of dichlorofluorescein (DCF), which was oxidized by ROS, at the emission wavelength of 535 nm (excitation wavelength: 485 nm) and using a microplate reader [7,31].

### 2.7. Antibacterial Activity of ABS@NPs

To estimate the antibacterial activity of ABS@NPs, a colony-counting method using *Staphylococcus aureus* ATCC 6538 (*S. aureus* ATCC 6538) was used. The bacteria were incubated in Luria–Bertani (LB) agar plates (BD Difco, Sparks, MD, USA) containing 1.5% agar at 37 °C. Before utilization, a single colony was inoculated in the LB broth and incubated overnight at 37 °C. This culture was subsequently diluted in the LB broth until the optical density (OD600) reached 0.1. Next, ABS@NPs (with 0 and 10 wt% ABS) were added to each culture to estimate the inhibition of bacterial growth over 24 h in an incubator. PBS and the ABS solution in DIW were used as negative control groups. Finally, after incubation, the suspensions were diluted to a factor of 10⁷, and 100 μL of each suspension was spread over an agar plate using glass beads. After overnight incubation, the number of colonies was counted to determine the antibacterial activity against *S. aureus* ATCC 6538 [42,43].

### 2.8. In Vitro Cytotoxicity of ABS@NPs

The biocompatibility of ABS@NPs was evaluated utilizing NIH 3T3 cells, which were incubated in the DMEM solution including 1% penicillin–streptomycin and 10% FBS. Cell viability analysis was performed using the 3-(4,5-dimethylthiazol-2-yl)-2,5-diphenyltetrazolium bromide (MTT) assay 24 h after the addition of ABS@NPs. First, the cells were cultured at a density of 10,000 cells/well in 96-well plates. After 24 h of culture, the cells were incubated with different concentrations of ABS@NPs (1–10 µM) and cultured at 37 °C for 24 h. Thereafter, a 1 mg/mL MTT solution was added to the medium, and DMEM solution (containing 1% penicillin–streptomycin and 10% FBS) was added to each well of the plate and incubated for 3 h. Finally, the medium was washed with PBS and DMSO-d6 was used to dissolve the purple formazan dye crystals. Using a microplate reader (BioTek, Winooski, VT, USA), the absorbance of the formazan generated by live cells was estimated at 570 nm. The percentage of viable cells was estimated using the following equation: [7,29,36,44]
(4)$$\mathrm{Cell}\;\mathrm{viability}\;\left(\%\right)=\left(\frac{\Delta A_{570}of\;test\;group}{\Delta A_{570}of\;control\;group}\right)\times 100$$

### 2.9. Statistical Analysis

The resulting data are presented as mean ± standard deviation values, and every experiment was performed in triplicate. Using Student’s t-test method, differences between experimental groups were compared. Statistical significance in all evaluations was confirmed at *p* < 0.05.

## 3. Results and Discussion

### 3.1. Characterization of ABS@NPs

ABS, a sesquiterpene alcohol, is a natural hydrophobic bioactive compound with various effects, including antioxidant and antibacterial activities [28,29,30,32,33]. However, its application in the bio-industry, such as in functional foods, pharmaceuticals, and cosmetics, is difficult owing to its low solubility and stability in the aqueous phase [28,29,30,31,37]. Therefore, it is necessary to develop a carrier that improves the stability and solubility of lipophilic ABS. In this study, polyglyceryl-4 caprate NPs with various ABS concentrations (ranging from 0 to 40 wt%) were produced by an encapsulation method. Polyglyceryl-4 caprate self-assembles in water to form stable NPs consisting of a hydrophilic shell and a hydrophobic core in which hydrophobic compounds can be encapsulated (Figure 1). The physicochemical properties, including PDI and size, of ABS@NPs did not differ significantly as the ABS concentration was varied (0–40 wt%) (Figure 2A,B). The ABS@NP 0 wt% (without ABS) exhibited a negative zeta potential of approximately −14.48 ± 0.92 mV, and the value tended to be the same as the concentration of ABS in the ABS@NPs was increased up to 10 wt%. However, in the ABS concentration range of 20 to 40 wt%, the surface zeta potential became increasingly negative as the loading content of ABS increased (Figure 2C). The encapsulation of ABS in the ABS@NPs was estimated using staining with a hydrophilic dye, methylene blue. Because ABS is colorless, the non-encapsulated part of the ABS@NPs cannot be observed well visually. With methylene blue added, the suspensions of ABS@NPs with up to 10 wt% ABS were clearly transparent, indicating that ABS was successfully encapsulated in the NPs up to a concentration of 10 wt%. However, at ABS concentrations above 10 wt.%, the suspensions became opaque, indicating the presence of unencapsulated ABS (Figure 2D). To quantify the encapsulation of ABS in the ABS@NPs, the ABS loading contents and efficiencies of ABS@NPs were analyzed via HPLC at different ABS concentrations, and the results are as follows: 5 wt% ABS@NPs: L.C. = 4.10, L.E. = 82.07, 10 wt% ABS@NPs: L.C. = 8.13, L.E. = 81.34, 20 wt% ABS@NPs: L.C. = 13.52, L.E. = 67.60, and 40 wt% ABS@NPs: L.C. = 15.17, L.E. = 37.92. Based on the experimental data, we concluded that 10 wt% is the optimal loading of ABS in ABS@NPs.

### 3.2. Long-Term Stability of The ABS@NPs

Particles of an oil phase in water are metastable, that is, they are thermodynamically unstable owing to their high energy level. Therefore, suspensions of oil particles dispersed in water using surfactants easily phase-separate through destabilization processes, such as precipitation, flocculation, and coalescence. Therefore, the long-term stability of ABS@NPs in DIW was analyzed at 25 °C by evaluating the changes in the NP size and PDI using DLS. The initial particle size (Figure 3A) and PDI (Figure 3B) of ABS@NPs indicated stability for a prolonged period, up to 112 days, without any notable change at ABS concentrations ranging from 0 to 10 wt%. In addition, all samples had a particle size of 300 nm or less and a PDI value of 0.3 or less. Thus, the results indicated that the fabricated ABS@NPs have remarkable long-term stability, and the system can be used as a successful platform for developing stable drug delivery systems.

### 3.3. In Situ DPPH Radical-Scavenging Activity and In Vitro Antioxidant Activity of the ABS @NPs

ABS is a stronger antioxidant than garlic [45]. However, maximizing the antioxidant activity of ABS in the aqueous phase is challenging because of its hydrophobic properties. Therefore, we fabricated ABS@NPs to maximize the ABS activity in an aqueous solution. To measure the antioxidant activity of ABS@NPs, an in situ DPPH antioxidant activity assay and an in vitro radical-scavenging activity assay were performed. First, using the DPPH radical-scavenging assay, the antioxidant activities of ABS@NPs and various control groups were evaluated (Figure 4A). DPPH, which is an organic nitrogen radical, absorbs at the wavelength of 515 nm in the visible range. When a solution of DPPH radicals reacts with an antioxidant, the color of the resulting solution shifts from violet to yellow because the DPPH receives electrons or hydrogen atoms from the radical scavenger, and its absorbance at 515 nm decreases. Thus, the percentage of scavenged DPPH radicals were determined based on the absorbance of the samples at 515 nm. ABS in DIW had little effect in scavenging DPPH radicals in comparison with the positive control of AA in DIW. In other words, this experiment showed that bio-applications are limited if alpha-bisabolol is not water-soluble through encapsulation. However, ABS@NPs exhibited more effective DPPH radical-scavenging activity than that of AA, and the antioxidant effect increased with the increase in the ABS concentration in the ABS@NPs from 5 to 10 wt%. In addition, in the case of ABS@NP 0 wt% (without ABS), there was little scavenging of DPPH radicals, suggesting that the solubilizer, polyglyceryl-4 caprate, had little effect on the antioxidant activity (Figure 4A). H2DCFDA, which is an ROS fluorescence indicator, diffuses through the cell membrane and is oxidized very quickly to highly fluorescent DCF, which fluoresces green in the presence of intracellular ROS, and its fluorescence intensity at the emission wavelength of 535 nm (excitation wavelength: 485 nm) increases [10]. Hence the in vitro antioxidant activity assay was conducted by analyzing H2DCFDA fluorescence after the addition of H_2_O_2_ in mouse NIH 3T3 fibroblast cells. H_2_O_2_, which is an oxidative stress agent, spontaneously formed a cellular environment, leading to the generation of ROS. The NIH 3T3 cells treated with H_2_O_2_ alone were used as the negative control group. As expected, the ROS levels of the cells treated with ABS@NP 10 wt% decreased considerably to approximately 15% as the ABS concentration increased from 1 to 1000 nM. However, the ROS levels of the cells treated with ABS in DIW and ABS@NP 0 wt% were reduced up to 49% and 56%, respectively. Therefore, we consider that these two samples have little effect on the ROS-scavenging activity in vitro (Figure 4B). These results indicate that the encapsulation of ABS enhances its antioxidant activity.

### 3.4. Antibacterial Activity of ABS@NPs

ABS has previously been reported to exhibit antibacterial activity [31,32]. However, owing to its hydrophobic properties, its antibacterial properties are not effectively manifested in the aqueous phase. In this study, we used a colony-counting method to analyze the antibacterial activity of ABS@NPs against *S. aureus* ATCC 6538. It was visually observed that the ABS@NPs significantly inhibited the growth of *S. aureus* compared to the control groups of PBS and ABS@NP 0 wt% (Figure 5A). The viability of *S. aureus* ATCC 6538 was reduced to 1 and 30% after treatment with ABS@NPs and ABS in DIW, respectively. Notably, these results show that the antibacterial effect of ABS increased considerably after encapsulation (Figure 5B), indicating that even a small amount of ABS exhibits considerably enhanced antibacterial activity.

### 3.5. In Vitro Cytotoxicity Activity of ABS @NPs

To assess the biocompatibility of ABS@NPs in NIH 3T3 fibroblast cells, an MTT assay was performed after treatment with ABS@NPs. There was almost no reduction in the viability of the NIH 3T3 cells. No toxic effects were observed in cells treated with ABS at concentrations ranging from 1 to 10 μM (Figure 6). We found that ABS@NPs did not show cytotoxic effects up to a concentration of 10 μM, and did not induce toxic effects in cells.

## 4. Conclusions

Bioactive compounds have various properties, such as antioxidant and antibacterial activities, and are widely used in the pharmaceutical, cosmetic, and food industries. ROS, which are highly reactive forms of intermediates, include H_2_O_2_, a superoxide radical (O_2_^−^), and a hydroxyl radical (OH^−^). They cause extreme damage to biomolecules in cells and mitochondria, thereby inducing inflammation and excessive oxidative stress in the body, resulting in various diseases, such as cancer and AD in humans. Therefore, compounds with antioxidant properties have become of great interest for the development of various bioproducts in modern urbanized society, where many factors, such as ultraviolet rays, fine dust, and blue light, induce the formation of active oxygen species. The antibacterial activity of natural substances is an important property for bioproduct development as it minimizes preservatives, such as formaldehyde, iodopropynyl butylcarbamate, and triclosan, which induce issues such as allergies and contact dermatitis in users. ABS, a sesquiterpene alcohol, is a natural hydrophobic bioactive compound with various effects, including antioxidant and antibacterial activities. However, its application in the bio-industry is difficult owing to its low solubility and stability in the aqueous phase. In this study, we demonstrated the successful and facile fabrication of ABS@NPs, which have potential antioxidant and antibacterial effects, using an emulsification method. The optimized ABS@NPs were thermodynamically stable and exhibited excellent stability for up to 112 days. Moreover, the improved aqueous solubility of ABS@NPs led to a remarkable increase in the bioavailability of ABS and induced improved effects of ABS compared to those of non-encapsulated ABS. We confirmed that ABS@NPs successfully scavenge ROS and kill bacteria without toxicity. The ABS@NPs are expected to be used as drug carriers with outstanding stability in various fields, including foods, cosmetics, and pharmaceuticals.

## Figures and Tables

**Figure 1 antioxidants-12-00207-f001:**
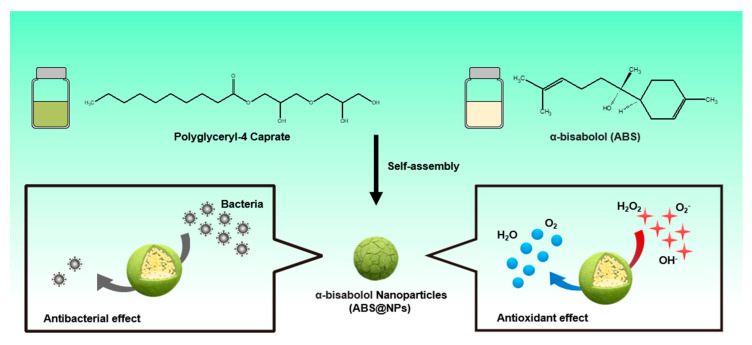
Schematic illustration of the preparation of α -bisabolol-loaded nanoparticles (ABS@NPs) and improved antioxidant and antibacterial effects.

**Figure 2 antioxidants-12-00207-f002:**
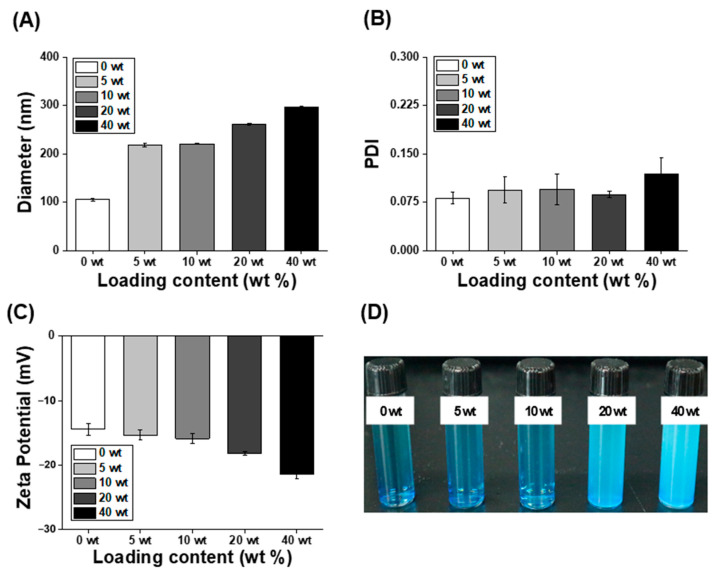
(**A**) Particle size, (**B**) polydispersity index (PDI), and (**C**) zeta potential of ABS@NPs with different ABS loadings in the range of 0 to 40 wt%. (**D**) Methylene blue staining test of the ABS@NPs with different ABS loadings.

**Figure 3 antioxidants-12-00207-f003:**
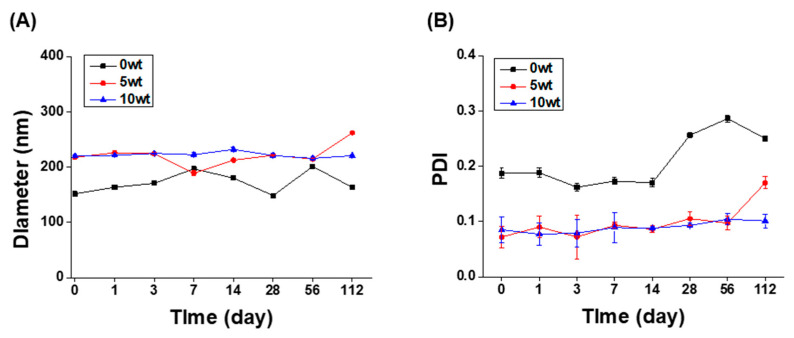
Long-term stability of ABS@NPs at ABS concentrations ranging from 0 to 40 wt%. Variation of the (**A**) particle size and (**B**) polydispersity index (PDI) of ABS@NPs over 112 days.

**Figure 4 antioxidants-12-00207-f004:**
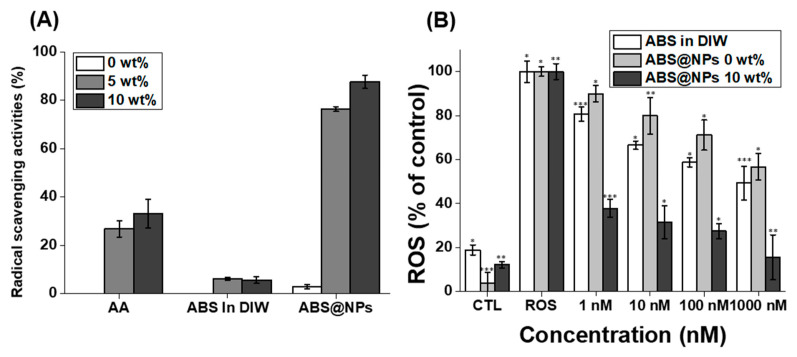
(**A**) Antioxidation activity of ABS@NPs with 0, 5, and 10 wt% ABS, compared to that of ascorbic acid (AA) in deionized water (DIW), assessed using the 2,2-diphenyl-1-picrylhydrazyl (DPPH) radical-scavenging assay. (**B**) Results of in vitro assays using the H2DCFDA assay kit. Antioxidation activity of ABS@NP 10 wt% at ABS concentrations ranging from 1 to 1000 nM; reactive oxygen species (ROS) and the control (CTL) groups represent the highest and lowest levels of ROS, respectively (* *p* < 0.05, ** *p* < 0.01, *** *p* < 0.005).

**Figure 5 antioxidants-12-00207-f005:**
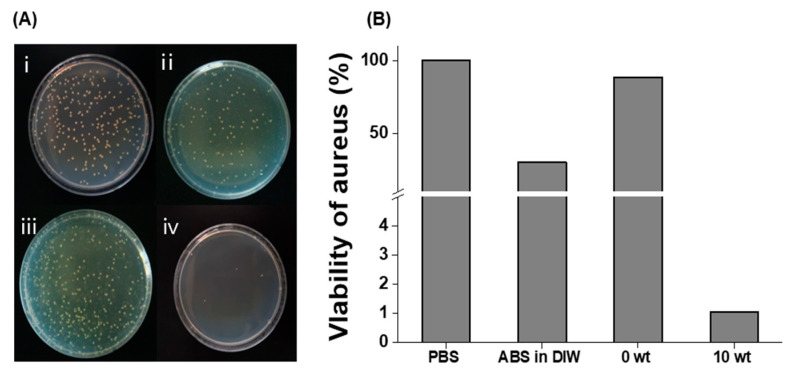
Antibacterial activity of ABS@NPs. (**A**) Photographs of staphylococcus aureus ATCC 6538 colonies in (i) phosphate-buffered saline, (ii) ABS in deionized water (DIW), (iii) ABS@NP 0 wt% in DIW, and (iv) ABS@NP 10 wt% in DIW. (**B**) Viability of S. aureus after treatment with phosphate-buffered saline (PBS), ABS in DIW, ABS@NP 0 wt% in DIW, and ABS@NP 10 wt% in DIW.

**Figure 6 antioxidants-12-00207-f006:**
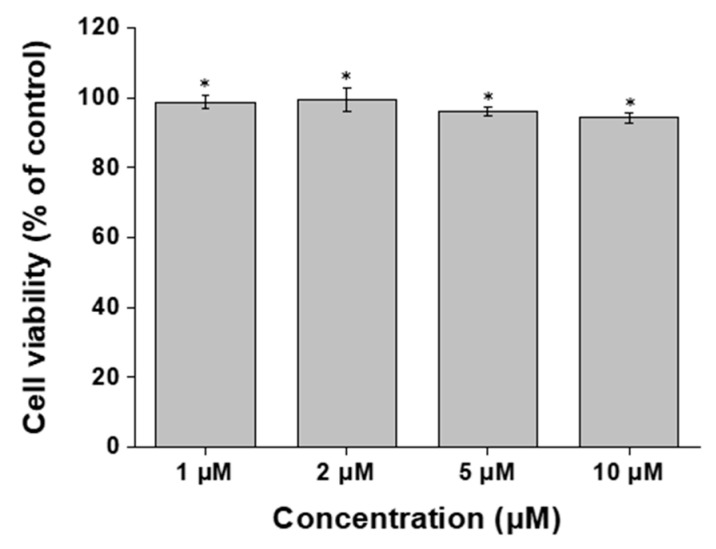
Cytotoxicity analysis of 10 wt% ABS@NPs at ABS concentrations in the range of 1 to 10 μM (* *p* < 0.05).

## Data Availability

Not applicable.

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
