# Peer review of "Facile Fabrication of α-Bisabolol Nanoparticles with Improved Antioxidant and Antibacterial Effects"

_antioxidants, 2023, doi:10.3390/antiox12010207_

Round 1

Reviewer 1 Report

The paper is well documented, the experimental data are meaningful and well interpreted and the conclusions reflect the capacity of NP to have bactericidal activity. The author have to present the HPLC conditions and a chromatogram. Also some TEM could provide supplementary proof of nanoparticle formation

Author Response

Thank you for your constructive comments. Each comment has been carefully considered, and responses are provided in a point-by-point manner.

Reviewer 2 Report

Dear Authors:

The manuscript "Facile fabrication of α-bisabolol nanoparticles with improved antioxidant and antibacterial effects" by Kim et al. has demonstrated that the successful and facile fabrication of ABS@NPs, which could have antioxidant and antibacterial effects, using an emulsification method. I have just a few suggestions.

1. The manuscript needs linguistic improvement.

2. Some references or background information is missing.  In introduction, please add more background information about ROS, which plays an important role in many diseases. especially mitochondria damage and AD development. It can emphasize the importance of your article. (Please cite:

1. An Epigenetic Role of Mitochondria in Cancer. Cells 2022, 11, 2518. https://doi.org/10.3390/cells11162518

2.    how far are we from truly understanding the pathogenesis of age-related dementia? Geroscience. 2022 Jun;44(3):1879-1883. doi: 10.1007/s11357-022-00591-7. 

3.  Mitochondrial mutations and mitoepigenetics: Focus on regulation of oxidative stress-induced responses in breast cancers. Semin Cancer Biol. 2022 Aug;83:556-569. doi: 10.1016/j.semcancer.2020.09.012.)

Best,

Author Response

(The authors gave the same response as above.)

Reviewer 3 Report

Kim et al have generated nanoparticles (NPs) that encapsulate alpha-bisabolol (ABS), a bioactive compound that acts both as an antioxidant and antibacterial agent. The authors convincingly show that they have successfully encapsulated ABS in the NPs (ABS@NPs) with increased sizes up to ~300 nm, that the PDI did not significantly change, but above 10% the particles show increasing negative zeta potential, indicating 10 wt% is the optimal concentration for further tests. This concentration also provided a transparent solution, indicating adequate solubility. The ABS@NPs were stable up to 112 days, and their NPPH antioxidant activity assays convincingly show ROS scavenging capacities. Their results from tests of in vitro ROS scavenging capacities with NIH 3T3 cells stimulated with H2O2 might be improved with primary data showing overall ROS production, which would help in understanding or predicting overall ROS scavenging capacities in vivo. Nonetheless, the ABS@NPs indeed show antioxidant behavior and importantly a robust antibacterial capacity against S. aureus, and were biocompatible at similar concentrations with the fibroblasts. Overall, the authors use a straightforward set of assays on their novel nanoparticle encapsulation system with ABS and their results suggest that the particles could be a useful therapeutic tool to suppress ROS while also providing antibacterial activities in biomedical applications. Some issues might be considered as follows:

Lines 275-282, the authors might better explain the use of increasing concentrations of ABS in Figure 4B, which tested the differences in ROS suppression in ABS in DIW, ABS@NP 0wt% vs. ABS@NP 10%. Why did they add additional ABS throughout the tests of ABS@NP? Why is this important to show in addition to the ABS supplied by the nanoparticle-encapsulated form? With a bit more explanation, as provided by the DPPH radical-scavenging assay that is nicely described, this will be easier for the reader to navigate.

Figure 4A, the authors might add "<1%" above the graphs indicating "0 wt%" for the negative control bar graphs above "AA" and "ABS In DIW", showing that the activities were negligible.

Figure 4 and Results/Discussion: What is the overall ROS production of the NIH 3T3 cells treated with H2O2, and how do those levels change with the ABS@NPs, other than just percentage of maximal response? Perhaps addition of a panel or a supplemental figure showing the fluorescence produced by the H2O2-treated cells and how this changed with the ABS@NPs would help, as these cells are not robust ROS producers as compared to innate immune cells (e.g.., stimulated neutrophils or macrophages).

Figure 6, did the authors test any other cell lines, in particular human cells, for example at least HEK 293 cells? A human model might add more confidence to their biocompatibility assay results on the mouse cells, or at least this should be suggested for future work.

Minor grammatical issues

Lines 47-50, the phrase "have become essential for the development of various bioproducts" requires editing, as the subject of "compounds with antioxidant properties" is not essential to the development of bioproducts. Perhaps change to "have become of great interest for the development of various bioproducts.....".

Line 62, after (ABS), the "a natural bioactive compound" has a "k" before the "a".

Author Response

(The authors gave the same response as above.)

Round 2

Reviewer 3 Report

The authors have addressed all concerns of this reviewer.